# The Surgical Histopathology of the Filum Terminale: Findings from a Large Series of Patients with Tethered Cord Syndrome

**DOI:** 10.3390/jcm13010006

**Published:** 2023-12-19

**Authors:** Hael Abdulrazeq, Owen P. Leary, Oliver Y. Tang, Helen Karimi, Abigail McElroy, Ziya Gokaslan, Michael Punsoni, John E. Donahue, Petra M. Klinge

**Affiliations:** 1Department of Neurosurgery, Warren Alpert Medical School of Brown University, Providence, RI 02903, USA; owen_leary@brown.edu (O.P.L.); jdonahue3@lifespan.org (J.E.D.);; 2Department of Neurosurgery, Rhode Island Hospital, Providence, RI 02903, USA; 3Department of Neurosurgery, University of Pittsburgh Medical Center, Pittsburgh, PA 15213, USA; oliver_tang@brown.edu (O.Y.T.);; 4Department of Neurosurgery, Tufts Medical School, Boston, MA 02111, USA; helenkarimi02@gmail.com; 5Laboratory Medicine, Department of Pathology, Rhode Island Hospital, Providence, RI 02903, USA

**Keywords:** filum terminale, spinal cord disorders, tethered cord release, tethered cord syndrome

## Abstract

This study investigated the prevalence of embryonic and connective tissue elements in the filum terminale (FT) of patients with tethered cord syndrome (TCS), examining both typical and pathological histology. The FT specimens from 288 patients who underwent spinal cord detethering from 2013 to 2021 were analyzed. The histopathological examination involved routine hematoxylin and eosin staining and specific immunohistochemistry when needed. The patient details were extracted from electronic medical records. The study found that 97.6% of the FT specimens had peripheral nerves, and 70.8% had regular ependymal cell linings. Other findings included ependymal cysts and canals, ganglion cells, neuropil, and prominent vascular features. Notably, 41% showed fatty infiltration, and 7.6% had dystrophic calcification. Inflammatory infiltrates, an underreported finding, were observed in 3.8% of the specimens. The research highlights peripheral nerves and ganglion cells as natural components of the FT, with ependymal cell overgrowth and other tissues potentially linked to TCS. Enlarged vessels may suggest venous congestion due to altered FT mechanics. The presence of lymphocytic infiltrations and calcifications provides new insights into structural changes and mechanical stress in the FT, contributing to our understanding of TCS pathology.

## 1. Introduction

Tethered cord syndrome (TCS) clinically presents with progressing back and leg pain, sensorimotor problems in the lower extremities, in addition to bladder and bowel dysfunction. This syndrome can occur in the pediatric and adult population. Magnetic resonance imaging (MRI) in TCS may show a low-lying conus, often associated with a thickened, shortened filum terminale (FT), with or without fatty infiltration seen in T1 weighted non-fat suppressed MRI. However, the presentation of TCS may be radiographically “occult”, considering that MRI may not reveal the histological and histopathological spectrum that resides within the FT [1,2,3,4]. Sectioning of the FT has been established as a surgical intervention with favorable results and symptom improvement in both the adult and pediatric tethered cord syndrome population [5].

The FT is a fibrovascular structure, anatomically “anchoring” the spinal cord to the distal thecal sac at the sacral 1–2 level (Figure 1). In addition to connective tissue, the FT histology has arbitrarily revealed both ependymal, ganglion, and glial cells, as well as arachnoid remnants, consistent with its embryonic origin [4,6,7,8], as it is formed during the secondary neurulation stage of development—the so-called “retrogressive differentiation” [9]. It is thus recognized that the FT is the remnant of an embryonic spinal cord [10]. The physiologic or functional role of the FT is largely unknown. Harmeier first described the histologic components of the cadaveric filum in 1933, which include populations of ependymal cells, neuroblasts, and corpora amylacea, with some glial cells, including oligodendrocytes, astrocytes, and microglial cells [11]. Selçuki et al. performed filum histology in stillborn human embryos, which revealed that various cell types, including ependymal and ganglion cells, exist within the FT [10]. The concept of a retained medullary cord as a result of defective secondary neurulation and cell apoptosis as described by Pang et al. proposed a mechanism for developing TCS due to altered structural and mechanical properties of the FT [12]. This is proposed, for example, in cases of fatty or thickened FT, which is thought to be caused by the abnormal presence of pre-adipose tissue or other embryologic remnants within the secondary neural tube [13,14,15]. This was also corroborated by the findings in human fetal histological examinations which revealed some differences to what is known of the adult cadaver FT, as these usually exhibit a lack of the peripheral nerve, glial, and ependymal tissues [3,16].

Fontes et al. were the first to describe the collagenous ultrastructure and integrity of the FT, which added to the concept of it serving as a physiological anchor to the spinal cord [17]. It has also been suggested that stress and strain injury to the conus medullaris and spinal cord may be an underlying pathological mechanism of TCS [2,4,18,19], which can lead to altered collagen structure due to a loss of elasticity and/or stiffness of the FT. More recently, we were able to indeed show disintegrated and pathologically “folded” collagen fibrils in a transmission electron microscope (TEM) study of the FT in patients who underwent surgery for TCS [4].

Given that the above histological knowledge mainly stems from cadavers, still-born embryonic humans, or other species [20], we present a large consecutive series of patients who underwent microsurgical resection of the FT presenting with TCS aiming to support the spectrum and prevalence of histological findings and observations within the FT. We aim to search for and describe biomarker and histological correlates that may be significant to possible structural and biomechanical alteration of the “diseased” FT. To support our observations, we include four case studies of TCS of increased vascularity and enlarged vasculature, lymphocytic pathologies, dystrophic calcifications, and ependymal proliferations in patients presenting with TCS.

## 2. Materials and Methods

Two hundred and eighty-eight consecutive patients presenting with TCS who underwent microsurgical resection of the FT at a single academic tertiary medical center between January 2013 and December 2021 were included in this study. These cases included both adult and pediatric cases in which the routine procedure consisted of a lumbar midline combined interlaminar/translaminar approach performed at the most proximal level of the FT with the removal of at least a 4 cm portion of the FT, which was then sent for pathologic analysis using a standard surgical technique recently described [21]. The patient cohort and the clinical symptoms and signs to support and substantiate the diagnosis of tethered cord syndrome are summarized in Table 1. All the patients were diagnosed and selected for surgical intervention based on the clinical judgement which considered the presence of symptoms and findings from at least two of the three components of the TCS symptom triad: back and/or leg pain, urinary dysfunction, and lower extremity motor or sensory symptoms. Neurocutaneous signs of spina bifida occulta, the presence of scoliosis, and imaging findings consistent with TCS such as thickened or fatty infiltrated FT and/or low-lying conus medullaris at or below L2 were supportive but not mandatory diagnostic elements. The standard preoperative imaging included a non-contrast MRI of the entire spine including T1- and T2-weighted sequences to highlight any fatty/lipomatous or thickened FT with or without a low-lying conus. The indication for surgery was further based on the following: (1) patients who exhibited clinical symptoms that were progressive and resistant to conservative treatment for ≥6 months, and (2) in whom non-neurogenic bladder dysfunction was ruled out by study-independent urologists. Urodynamic studies were not mandatory due to low sensitivity and specificity. All the specimens were reviewed by a board-certified neuropathologist. The IRB approval was obtained for retrospective analysis of filum pathology (IRB #787945). The findings constitute the pathology report where the neuropathologist noted and commented on those observations during the routine examination of the surgically obtained FT specimen.

Standard hematoxylin and eosin (H and E) stain was used to identify the findings mentioned below. Additionally, special stains and immunohistochemistry as indicated by the neuropathologist were added to select cases. The surgical specimen was evaluated for neuronal, glial as well as arachnoid cells, as well as reactive or degenerative structures such as corpora amylacea, Rosenthal fibers, and psammoma bodies. The fibrovascular tissue component was analyzed for the spectrum of cells known to reside in connective tissues, i.e., lymphocytes, mast cells, macrophages, and melanocytes.

To perform a literature review and identify prior investigations of these findings within the filum terminale, we used the following keywords that appeared in the pathology reports to perform a PubMed search: ependymal cells, ganglion cells, peripheral nerve, nerve twigs, lymphocytes, macrophages, mast cells, giant cells, lymph follicles, arteriovenous malformation, venous angioma, hemangioma, calcification, meningoepithelial cells, elastin fibers, corpora amylacea, psammoma bodies, Rosenthal fibers, and melanocytes. Reports that included these keywords and confirmed the presence of these findings within the specimen were included in the review.

## 3. Results

### 3.1. Results

The mean age of patients was 29.0 years (SD = 21.3, range = 0.5–72.7). Of note, 77 patients (26.8%) were younger than 10 years old, while an additional 41 patients (14.2%) were aged 10–18. Overall, 202 patients (70.1%) were female. On average, patients presented with 2 of the 3 components of the TCS triad regardless of imaging findings. The distribution of patients exhibiting each of the diagnostic sub-components is shown in Table 1. The most prevalent clinical presentation was urological symptoms in 182 (63%) patients, followed by 212 (73.6%) presenting with lower extremity neurological symptoms and signs, and 203 individuals (70.5%) with lower extremity and/or back pain. Three months post-surgery, 185 patients (64.2%) returned for postoperative follow-up at 3 months and the proportion of patients with the TCS triad—back pain, neurological, or urological symptoms—decreased to 62.7%, 36.8%, and 34.6%, respectively. At 12 months post-surgery, a total of 163 (56.6%) patients who followed up also showed reduced frequency of individuals reporting back pain (63.2%), neurological (51.5%), or urological (49.1%) symptoms when compared to the above preoperative prevalence, supporting a reversible TCS or symptom triad [5].

The histopathological findings and distribution by age in our cohort of 288 patients are summarized in Table 2. One hundred and eighteen patients (41%) exhibited various extents of fatty FT infiltration on histology. Of note, in only 147/288 (51.0%) patients was this seen on the sagittal and axial T1 non-fat suppressed MRI sequences. Some cases with ependymal cells also demonstrated an entirely patent ependymal canal (5.9%), ependymal cysts (3.1%), and one case showed extensive ependymal “proliferation” (Case #3). With regards to inflammatory cellular pathologies, eleven specimens (3.8%) contained lymphocytes, macrophages, mast cells, or giant cells, and in one patient the unusual finding of a secondary lymph follicle was described (Case #1). Increased vascularity and density of enlarged capillaries or venules compatible with the impression of “venous congestion” was found in 36 (12.5%) patients, to the extent that an associated venous angioma or arteriovenous malformation was suspected but not confirmed in the pathological specimen of nine patients (3.1%) (Case #2). Additionally, a frequent finding included dystrophic calcifications within the FT (7.6%), and in a few patients, we also noticed a variable amount of calcium deposits in the pial wall of the FT.

### 3.2. Case Illustrations

#### 3.2.1. Case #1: Lymphoid Proliferation Suggestive of a Secondary Lymph Follicle

A 34-year-old female presents to the neurosurgical clinic for evaluation of tethered cord syndrome. She has a past medical history of asthma, hyperlipidemia, sleep apnea, endometriosis, depression, gastroesophageal reflux disease, and migraines. She endorses chronic low back pain which started at 11 years of age. The pain is worse in ambulation and causes weakness and exhaustion if she walks more than 30 feet. She also has pain in a patchy distribution in her legs, feet, and buttocks. Additionally, she endorses feeling numb in her legs which is worse when she lies flat. She also reports occasional urinary incontinence. An MRI of the Lumbar spine showed evidence of a thickened filum with a conus terminating at the L1-2 disc space (Figure 2A). MRI of the rest of her neural axis did not reveal any abnormalities.

She underwent lumbar laminectomy for resection of filum terminale in the tethered cord. The H and E-stained sections were notable for small lymphoid aggregates infiltrating the fibro-adipose tissue structures of FT. The small lymphoid aggregates form secondary lymphoid follicles with well-demarcated and polarized germinal centers that become better visualized in the deeper sections used for immunohistochemistry studies (Figure 2B). However, the specimen was noted to contain foci of secondary B-cell follicles with germinal centers with normal phenotype (Figure 2C). Immunohistochemistry revealed CD20+, PAX5+ B-lymphoid follicles with germinal centers with normal bcl6+, CD10 weak+, bcl2 negative immunophenotype, and high proliferation rates, as highlighted by their immunoreactivity to the MIB1 antibody and Ki-67 antigen. These germinal centers are surrounded by CD20+, bcl2+ cuffs of small B-lymphoid cells with low proliferation rates (<5%). Adjacent to the B-lymphoid follicles is a small subset of small CD3+, bcl2+ T-cells with low proliferation rates (<15%), and there are a few scattered TdT+, CD34 negative nuclei located away from the B-lymphoid aggregates, favored representing a few reactive hematogones. (Figure 2D). A hematopathologist reviewed this study and interpreted it as a mixed reactive lymph follicle: immunohistochemical stain of CD20 markers demonstrates the predominance of B-cells within the lymphoid follicles most consistent with hematogones. Hematogones typically comprise of B-lineage lymphoid precursor cell population in the bone marrow which may simulate acute lymphoblastic leukemia or lymphoma. A full spine MRI with and without contrast was obtained at 3 months post-operatively which was unremarkable. The patient was also referred to a hematologist for further workup which did not reveal any hematological disorder.

#### 3.2.2. Case #2: Vascularity

This is a case of a 53-year-old male with prior lumbar 4–5 transforaminal lumbar interbody fusion who presented with a new onset of lower lumbosacral back pain, bilateral leg pain and numbness, and urinary incontinence and urgency. On the physical exam, his right leg strength is 4 out of 5 in the hip flexion and knee extension. He is numb to touch and has increased tone bilaterally. His reflexes are 1+ bilaterally, with no foot clonus. He had an MRI of the lumbar spine which revealed a thickened filum terminale and conus terminating at the L2/L3 disc space (Figure 3A). The patient then underwent a lumbar 2–4 laminectomy for resection of filum terminale with no complications.

Histopathological analysis revealed fragments of fibrovascular and adipose tissue with embedded and adjacent ependymal cells and peripheral nerve twigs, and notably large, thick-walled, congested blood vessels (Figure 3B). Elastin stain showed a blood vessel with an internal elastic lamina, confirming its arterial origin (Figure 3C). This was interpreted as an arteriovenous malformation.

#### 3.2.3. Case #3: Ependymal Cell Proliferation

A 61-year-old female presented with back pain, right leg weakness, and increased tone. She also complains of urge incontinence and underwent urodynamic studies which revealed a neurogenic bladder. She had an MRI of her spine which revealed evidence of a thickened filum with conus terminating at L2 (Figure 4A). As a result, we proceeded with a lumbar laminectomy for spinal cord detethering and resection of the filum terminale.

On histological analysis, the H and E stain revealed fatty infiltration of the entire filum terminale segment with focal hypercellularity and proliferation of ependymal cell types, with clean margins around the hypercellular area (Figure 4B). A higher-power image of the incidental ependymal cell proliferation is shown in Figure 4C. To further characterize the cell population in this specimen, glial fibrillary acidic protein (GFAP) immunohistochemical staining was performed. This demonstrated diffuse positivity within the hypercellular area. The proliferative index on Ki-67 immunostain was less than 1% (Figure 4D). The final interpretation was a focal ependymal cell infiltration and proliferation within the FT.

#### 3.2.4. Case #4: Focal Calcifications

A 28-year-old female presented to the neurosurgical clinic with a long-standing history of pain in her back and joints since her teenage years. She had also suffered from fatigue and was diagnosed with fibromyalgia in the past as well as chronic fatigue syndrome, then later was also diagnosed with dysautonomia and Ehlers-Danlos-Syndrome (EDS). She described pain in her back and legs which worsens with ambulation, and LE cramps that occur at night. She also reported urinary issues and had had a urodynamic study done that showed small bladder capacity. She had an MRI of her lumbar spine which demonstrated the presence of a conus medullaris tip between the L1-2 disc space and axial T1 scans notable for the hyperintense signal at the conus-filum transition zone and the L2-3 level suggestive of thickened filum terminale (Figure 5A). She underwent a lumbar laminectomy for resection of filum terminale, and the histopathological examination showed FT (Figure 5D) consistent with fragments of fibrovascular connective tissue with embedded and adjacent ependymal cells, peripheral nerves, ganglion cells, and large thin-walled blood vessels with notably large areas of focal calcifications alongside the blood vessels (Figure 5E). Immunohistochemistry staining revealed scattered CD45-positive lymphocytes.

## 4. Discussion

The FT is a fibrovascular band that serves as an anatomical anchor of the spinal cord to the distal thecal sac at the sacrum and acts as a physiologic “buffer” of biomechanical forces to the spinal cord. The association of the histological structure of the FT and its pathological capacity in TCS in humans will remain hypothetical since the FT surgical specimen lacks healthy surgical controls. With our large consecutive series of the histology of the surgical FT in TCS, we report the prevalence and spectrum of embryonic and connective tissue elements within the FT. While some of the structures, such as peripheral nerves, are normally present in the FT, our data proposes that either the excess of any tissue congenitally residing in the FT and/or acquired structural and “dystrophic” alteration of the fibrovascular tissue (i.e., venous congestion, calcification, or lymphocytic infiltration) possibly secondary to biomechanical strains may be relevant histological and pathological findings in TCS. We highlight the above findings within our cohort and discuss their possible correlation with the presentation of TCS.

### 4.1. The Implication of Peripheral Nerves and Fat within the FT

Previous studies have concluded that the occasional observation of peripheral nerve fibers found in surgical specimens of the FT was without any functional role [22]. Peripheral nerve structures in our series, however, were embedded within the fibrovascular tissue in almost all individuals. This supports the notion that these may be physiological structures of the FT as shown in an overlapping study of 56 patients where we provided electron microscopic and electrophysiological evidence that those are functional nociceptors and mechanoreceptor elements that may play an active role in the anchoring of the spinal cord within the spinal canal, providing “feedback” sensors for spinal alignment and spinal cord motion [22].

Fatty infiltration of the FT, on the contrary, has always been considered the hallmark pathology seen in TCS, either as embryonic “left-over” tissue from the failure of embryonic disjunction of the fetal spinal cord tissue, or an acquired age-related fatty degeneration [5]. It has been traditionally trusted that the MRI will diagnose or verify this pathology and, therefore, an MRI finding of a “fatty filum” was coined a diagnostic marker of TCS [23]. Our case studies, however, question the sensitivity of the MRI to detect or serve as a reliable marker for FT pathology. While only 40% of our patients had fatty tissue of various degrees within the filum terminale, this questions its relevance to the clinical syndrome. Seldon et al. have correlated the presence versus absence of fat in the filum with urinary symptoms and the outcome of urinary function and found the presence of fat to be positively linked to improved urinary outcome post-FT sectioning in a pediatric series of patients [24].

Regarding age distribution of tissue components, it was intriguing that structures of a “dystrophic” nature, such as corpora amylacea, Rosenthal fibers, psammoma bodies, and calcifications were mostly observed in adults, and we suspect that those in fact might be acquired, and age-related changes. Psammoma bodies, for example, have occasionally been described at the junction between the filum internum and externum [17,25]. Psammoma bodies are typically seen in arachnoid cells and suggest dystrophic meningothelial/arachnoid changes.

### 4.2. Ependymal Proliferation and Vascularity within the FT

The presented histological cases of TCS support that the predominance of certain cell types that normally reside in the FT may carry relevance to its pathological alteration and disease mechanism regardless of whether it is acquired or congenital.

Ependymal cells, for example, are commonly present in the FT when analyzed in fetal specimens, and though these can be found in specimens obtained from patients with TCS, this is less common compared to fetal specimens. This suggests that these fila contain an embryonic remnant of the fetal spinal cord and that the observed patent ependymal canals and ependymal cysts in less than 5% of our cohort may present physiological embryonic structures of the FT. The observed ependymal cell proliferation within the FT, as shown in our case example, can be a result of unusual ependymal infiltration due to a reactive process. In TCS, altered FT biomechanical stresses may cause these cells to progress to abundant proliferation, particularly since histologic cadaver studies have found that ependymal cells more frequently can be found in the proximal one-third of the filum and within islets of cells within the rest of the filum, which suggests why ependymomas tend to occur within various locations of the filum [26]. Myxopapillary ependymomas are a well-described entity and are thought to arise from the ependymal cells which can be normally present within the filum. This is the most common category of tumors described within the filum. Histologically, they are characterized by abundant mucin production and the appearance of papillary zones, rosettes, and pseudorosettes [27]. Whether or not these ependymal proliferations are precursors to myxopapillary ependymomas is speculative.

As much as vascular anomalies and malformations have been associated with a diseased FT [10,24,28,29,30,31,32], our findings of increased vascularity and enlarged diameter of capillaries and venules in more than 1/10 patients, both adults and children, so far have only been observed and described in the pediatric patients with symptoms of tethered cord [10,29]. The venous anatomy in the FT is more variable than that of the arterial supply, as the veins have been shown to have more variety in their caliber from subject to subject in cadaveric studies [28]. As such, the pathophysiology of tethering in some patients is speculated to be associated with microvascular compression in the setting of “filum tethering” and anomalies within the vascular system in the filum. More prominent vessels within and adjacent to the FT may comprise a histological biomarker of venous hypertension and congestion suggestive of a mechanism of venous pooling from ongoing tethering in patients with an otherwise unremarkable MRI, as suggested by Selkuci in his pediatric series [10]. The mechanism of venous congestion and pooling might be equivalent to the pathology of “varicose” veins, given that the FT is involved in the physiological venous drainage of the spinal cord [28]. In addition, a cadaver study of the vascular supply of the FT has shown evidence that the FT is involved in the venous drainage of the spinal cord, which supports the hypothesis that an anomalous FT might exhibit an increased number and diameter of venous and capillary structures, described as “increased vascularity” and proposed to be involved in the mechanism of a pediatric series presenting with TCS [10]. Of note, in nine patients in our cohort, including the presented case, the above vascular changes were so prominent, that it raised the suspicion of a vascular malformation of the FT, which is relatively rare, and the diagnosis can often be missed due to inability to detect these lesions on MRI. Though vascular malformations may be associated with radiographic findings of tethered cord with a low-lying conus, especially in cases with an associated fatty filum, some can be present in patients with a radiographically “normal” appearing conus or filum, particularly since these malformations may be difficult to detect with conventional imaging techniques. These vascular anomalies may not only disturb the normal venous drainage of this region of the conus and spinal cord leading to ischemic changes and myelopathy but may also cause altered elastic properties of the FT, promoting a symptomatic tethered cord. Additionally, shunting between the arterial and venous systems can lead to increased venous congestion resulting in spinal cord ischemia and myelopathy [33,34]. The presence of vascular malformations has been thought to produce symptoms due to the abnormalities in the physiology of venous drainage, and the resulting increased venous congestion leading to spinal cord ischemia and neurologic dysfunction [33,34].

However, the development of vascular abnormalities may also be the result of the presence of congenital findings such as a lipoma, which can increase the blood flow to the filum and make these vascular findings symptomatic [32,35,36,37].

### 4.3. Inflammatory Cells and Calcifications

To our knowledge, there have not been reports examining lymphatic or other inflammatory cell types within the FT in surgical specimens from patients manifesting with TCS. The occurrence of inflammatory cells including lymphocytes in the FT, as seen in 4% of our series, may be a sign of a coexisting local inflammatory or reactive process within the surrounding cauda equina or arachnoid layer, which in turn can cause nerve irritation and lower extremity symptoms, as well as urologic issues. The implication of the formation of lymphoid aggregates with secondary B-lymphoid follicles found in our patient remains unclear. Based on the morphologic, immunophenotypic, and molecular studies which did not detect the presence of a clonal immunoglobulin heavy chain gene rearrangements, a reactive process in this location is favored which may have been driven by an infectious (i.e., viral, bacterial, etc.) or inflammatory process. Malignant lymphoid proliferations have been rarely described within the region of the cauda equina [38,39], and in such cases, further hematological and oncological workup and MRI imaging of the neuroaxis may be warranted. Even though, in our case, no indication of a hematologic condition was found, we propose that the findings of inflammation and lymphocyte proliferation in the surgical specimen found in patients with TCS may warrant further workup as done in this patient. Not highlighted as cases part of the cases above, however, we found unexpected accumulation of mast cells in two of the pediatric cases (Table 2) as evidenced by CD 117 positivity. Of note, there was no clinical evidence or further workup for any widespread inflammatory or systemic co-morbidities in our patients, except in our EDS patient population. We were intrigued by the finding of positive CD 117+ stained “active” mast cells observed in some fila. Almost 20% of our patients were diagnosed with Ehlers–Danlos syndrome, which has a known association with mast cell activation syndrome [40,41]. Even though the sample size on mast cell occurrence is too small to correlate with the incidence within EDS, this might suggest further studies. Even though the findings of inflammatory or reactive processes within our patients did not trigger any further workup, except in the presented case example, or imaging given the novelty of those observations, they may still imply further workup or imaging for underlying pathological processes within the rest of the central nervous system, considering the presence of any relevant co-morbidities.

Calcifications have not yet been reported in the FT. These were found in no less than 7.6% of the patients included in this series. We hypothesize that the presence of calcifications within the FT is equivalent to what is found in mechanically overused tendons, pathologically consistent with tendonitis [42]. As such, this might indicate mechanical overuse of the FT in patients with TCS as a mechanism of the disease process since the FT is acknowledged as a biomechanical “buffer” of our spinal cord, similar to the mechanisms of muscle tendons [4]. Pathologic calcification in cauda equina elements has also been rarely described, and in some extreme cases may present as arachnoiditis ossificans, in which the arachnoid and dura develop calcifications. This is thought to be the result of a chronic inflammatory process, resulting from infection, trauma, or subarachnoid hemorrhage, and is the result of the progression of ongoing arachnoiditis [43,44]. It is therefore possible that the FT may develop a similar reactive deposition of calcium when it is inflamed, which may cause patients to develop symptoms of TCS. In fact, the finding of CD45 immunopositive lymphocytes along with the calcifications found in the patient presented here may support this nature.

## 5. Limitations

Though our study is certainly limited by the lack of mechanistic proof that any of the findings are causative to the patients’ symptoms, our conclusions are based on what little knowledge exists on “normal” filum anatomy, and the obvious limitations on mechanistic studies to prove the concept. We did not provide any correlation of the histological and structural findings with symptom severity or clinical outcome post-filum sectioning which would probably provide a more robust correlation to a true pathology in the FT associated with TCS. However, TCS lacks a grading system of system severity [5]. Furthermore, any correlation to outcome in the absence of a reliable MRI biomarker and post-surgical tissue markers has no clinical or predictive relevance and/or clinical implications.

## 6. Conclusions

This case series represents a histopathologic review of the FT in the largest reported consecutive series of TCS patients in which a significant segment of the FT was removed via the surgeon’s standard technique. Novel and underreported findings included vascular and inflammatory histology in the surgical FT. Calcifications possibly support the acquired pathophysiological concept of biomechanical stress or strain to the spinal cord in TCS, adding to the damage of the spinal cord which might particularly be appreciated in adult patients over time or with traumatic stressors.

## Figures and Tables

**Figure 1 jcm-13-00006-f001:**
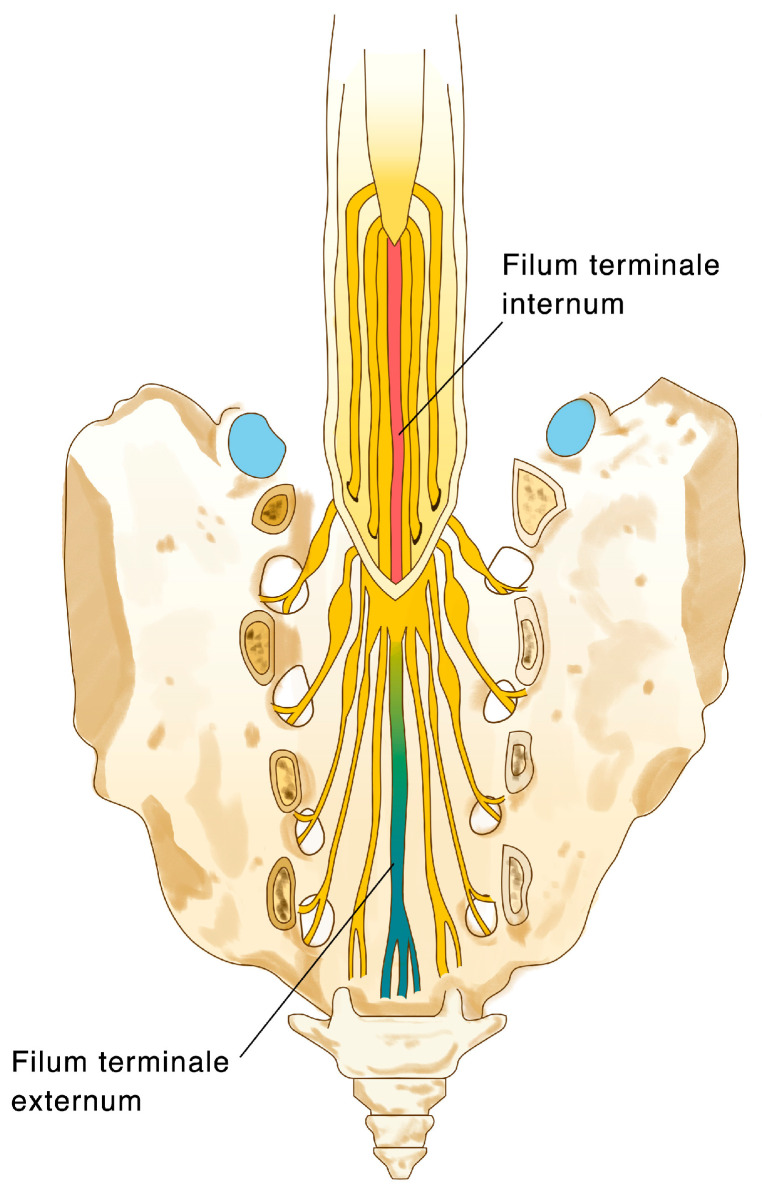
Illustration of the anatomy of the filum terminale, and its transition from the filum terminale internum (red), to the filum terminale externum (blue) at S1-2.

**Figure 2 jcm-13-00006-f002:**
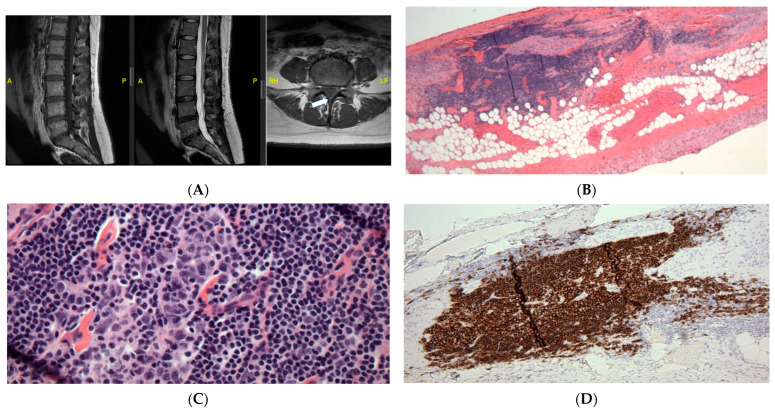
MRI (**A**) and histologic (**B**,**C**) findings of the lumbar spine. (**A**) Sagittal T1 (left) and T2 (middle) sequences show the conus terminating at the L1-2 disc space, and an axial T1 sequence (right) reveals a hyperintense signal of the filum (arrow), suggestive of thickened FT. (A: Anterior, P: Posterior, RH: Right, LF: Left). (**B**) Hematoxylin and eosin (HE) stain shows FT with prominent lymphoid follicles and focal adipose tissue, ×40. (**C**) The HE stain reveals a lymphoid cell population with focal germinal centers, ×400. (**D**) CD20 immunohistochemical stain of FT demonstrates the predominance of B-cells within the lymphoid follicles most consistent with hematogones, ×100.

**Figure 3 jcm-13-00006-f003:**
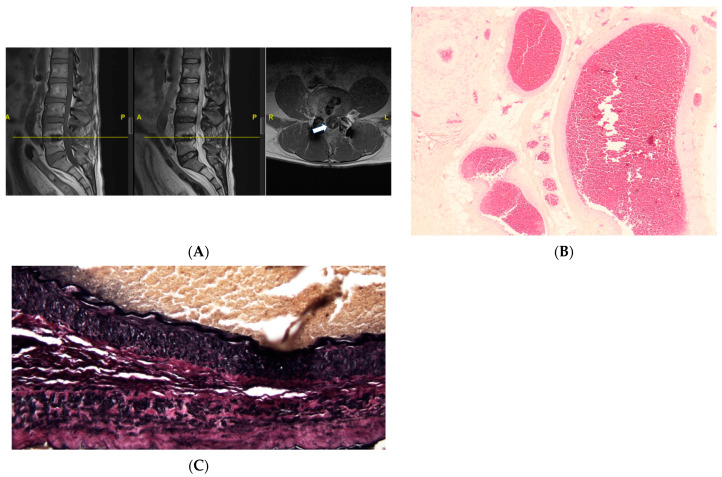
MRI (**A**) and histopathological examination of the FT (**B**,**C**). (**A**) Sagittal T1 (left), T2 (middle), and axial T1 sequence (right) reveal a hyperintense signal of the filum (arrow), suggestive of thickened FT. (A: Anterior, P: Posterior, R: Right, L: Left). (**B**) Fatty FT with several large, thick-walled, congested blood vessels, suggestive of a venous congestion. The H and E stain, ×40 (**C**) Low power of an elastic stain demonstrating a blood vessel with an internal elastic lamina, confirming its arterial origin. This suggests signs of venous congestion, ×40.

**Figure 4 jcm-13-00006-f004:**
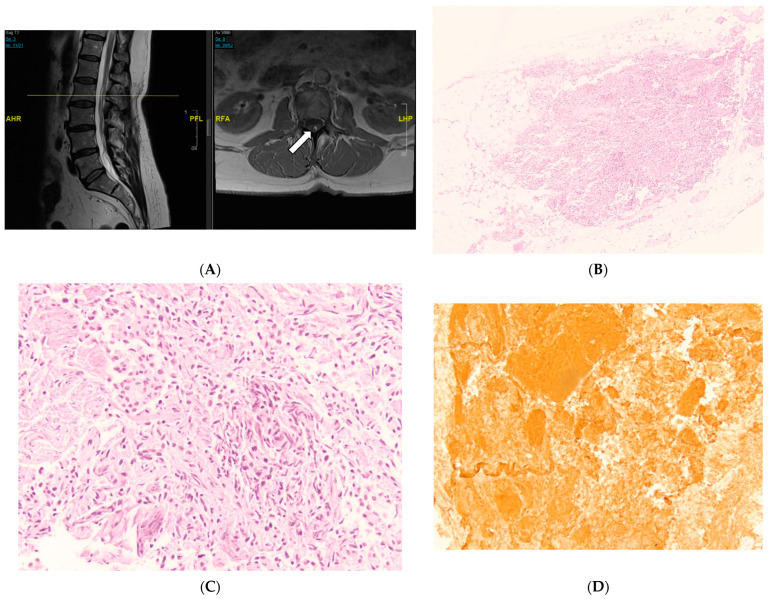
MRI (**A**) and histological examination (**B**–**D**) of the FT. (**A**) MRI of the lumbar spine with conus terminating at L1-2 disc level, and axial sequence reveals a T1 hyperintense signal suggestive of thickened filum (arrow). (AHR: Anterior. PFL: Posterior. RFA: Right. LHP: Left.) (**B**) Fatty filum terminale with focal, incidental ependymal cell proliferation. H&E stain, ×40. (**C**) Higher power of incidental ependymal cell proliferation. An ependymoma cannot be completely ruled out, but no pathognomonic features are present. H&E stain, ×200. (**D**) GFAP immunohistochemical stain demonstrating diffuse positivity within the ependymal cell proliferation. The proliferative index on Ki-67 immunostain is less than 1%, ×200.

**Figure 5 jcm-13-00006-f005:**
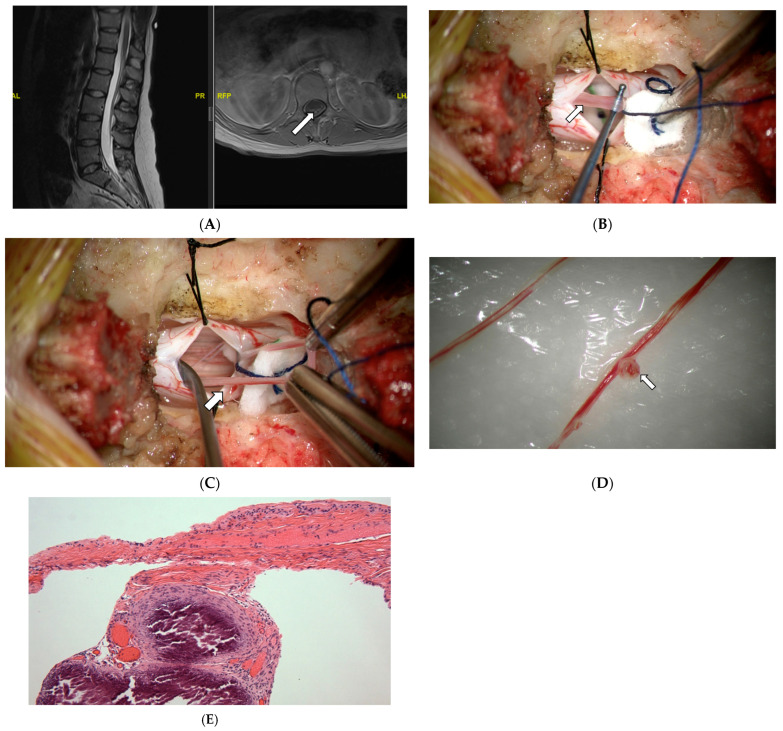
MRI (**A**), intraoperative exploration (**B**,**C**), macroscopic (**D**), and microscopic (**E**) examination of the FT. (**A**) MRI of the lumbar spine showing the tip of conus medullaris at the L1-2 disc space, and axial T1 scans notable for the hyperintense signal at the conus-filum transition zone suggestive of thickened filum terminale (arrow). (AL: Anterior. PL: Posterior. RFP: Right. LHP: Left). (**B**) The intraoperative picture shows the surgical approach and dural opening after performing a single level laminotomy, with the calcified filum (arrow) lifted using a nerve hook instrument. (**C**) The intraoperative picture demonstrates the surgical approach with dural opening, filum isolation, and resection of a larger filum portion (arrow). (**D**) gross pathological specimen with evidence of large focal calcification (arrow). (**E**) Filum terminale with focal calcifications alongside thin-walled blood vessels. HE stain, ×40.

**Table 1 jcm-13-00006-t001:** Presenting characteristics of included patients.

Characteristic (Total N = 288)	N (%) or Mean (SD)
Age at Surgery (years)	29.0 (±21.3)
Adult Patients (>18 years)	170 (59.0%)
Pediatric Patients (10–18 years)	41 (14.2%)
Pediatric Patients (<10 years)	77 (26.8%)
Gender (Female)	202 (70.1%)
Diagnosis with Tethered Cord Syndrome	288 (100%)
Mean # of TCS Triad Symptoms	2.0 (±1.2)
Back and/or Leg Pain	203 (70.5%)
Neurological Signs and Symptoms	212 (73.6%)
Urological Symptoms	182 (63.2%)
Patients at 3-month follow-up	185 (64.2%)
Back and/or Leg Pain	116 (62.7%)
Neurological Signs and Symptoms	68 (26.8%)
Urological Symptoms	64 (34.6%)
Patients at 12-month follow-up	163 (56.6%)
Back and/or Leg Pain	59 (63.2%)
Neurological Signs and Symptoms	84 (51.5%)
Urological Symptoms	80 (49.1%)

**Table 2 jcm-13-00006-t002:** Summary of histopathologic findings within the filum terminale of surgical specimens from tethered cord syndrome patients and their age distribution.

Histopathologic Finding (Total N = 288)	N (%)	Age Distribution (# of Patients, Percentage Out of Entire Sample)		
		Pediatric (<10 Years)	Pediatric (10–18 Years)	Adult Patients (>18 Years)
Peripheral Nerves	281 (97.6%)	81 (28.12%)	40 (13.89%)	160 (55.56%)
Ependymal Cells	204 (70.8%)	49 (17.01%)	30 (10.42%)	125 (43.4%)
Fatty infiltration	118 (41%)	47 (16.32%)	16 (5.56%)	55 (19.1%)
Ganglion Cells	111 (38.5%)	29 (10.07%)	17 (5.9%)	65 (22.57%)
Neuropil	44 (15.3%)	17 (5.9%)	2 (0.69%)	25 (8.68%)
Vascular findings	36 (12.5%)	11 (3.82%)	4 (1.39%)	21 (7.29%)
Meningothelial Cells	14 (4.9%)	5 (1.74%)	1 (0.35%)	8 (2.78%)
Inflammatory cells	11 (3.8%)	2 (0.69%)	1 (0.35%)	8 (2.78%)
Elastin Fibers	9 (3.1%)	4 (1.39%)	1 (0.35%)	4 (1.39%)
Corpora Amylacea	6 (2.1%)	0 (0.0%)	0 (0.0%)	6 (2.08%)
Psammoma bodies	3 (1.0%)	0 (0.0%)	0 (0.0%)	3 (1.04%)
Rosenthal fibers	2 (0.07%)	0 (0.0%)	0 (0.0%)	2 (0.69%)
Melanocytes	1 (0.03%)	1 (0.35%)	0 (0.0%)	0 (0.0%)

## Data Availability

The data presented in this study are available on request from the corresponding author. The data are not publicly available due to privacy.

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
