# Peer review of "The Surgical Histopathology of the Filum Terminale: Findings from a Large Series of Patients with Tethered Cord Syndrome"

_jcm, 2023, doi:10.3390/jcm13010006_

Round 1

Reviewer 1 Report

Comments and Suggestions for Authors

In the manuscript entitled The Surgical Histopathology of the Filum Terminale: Findings from a Large Series of Patients with Tethered Cord Sydrome authors investigated the prevalence of embryonic and connective tissue elements in the filum terminale (FT) of a large series of patients with tethered cord syndrome (TCS), examining both typical and pathological histology. Their aim was to search for and describe biomarker and histological correlates that may be significant to possible structural and biomechanical alteration of the diseasedFT. They conclude by adding novel and underreported findings including vascular and inflammatory histology in the surgical FT.

First, we would like to congratulate with the authors for their interesting work and efforts.

Please, accept our following few criticisms:

-     The definitions of filum terminale reported in the “Introduction” and “Discussion” sections are different; please use the same definition;

-     As authors rightly stated in the text, all the considerations about the role of  histopathological findings of filum terminale in the etiopathogenesis of tethered cord syndrome can be only hypothesized as a normal surgical specimen control is not available.

-     The novel inflammatory findings noted in only 4% among overall series should be correlated with other factors involved in systemic conditions; were these patients affected by other comorbidities? Have Inflammatory indexes of systemic diseases been investigated?

Albeit some of the findings reported in the present manuscript only confirm the data of the pertinent literature, while the novel ones  described are only hypothetical, we consider this paper useful for increasing the knowledge and understanding on this topic.

Author Response

Dear Reviewer and Editorial Team of JCM,

The authors greatly appreciate the feedback and comments provided by the reviewer, and have addressed these comments with our submitted manuscript, and provide the following point-by-point response to the reviews:

  • The definitions have been edited of the filum to be concordant between the introduction and discussion. Specifically, the definition in the discussion has been edited to match the introduction. (Lines 272-273)
  • The comment is noted as we do believe any correlations between histopathology and pathophysiology of tethered cord syndrome is only hypothetical given the limitations of this study.
  • We clarify in the discussion under the inflammatory section, that no other inflammatory or relevant systemic co-morbidities were reported in the patients with inflammatory cells in their fila, and that, other than the case illustration which we highlighted, none of the patients in this cohort received further work up for widespread inflammatory findings, or further lab work up or inflammatory markers, given the work up for these findings has not been a standard of practice, in the absence of other signs or symptoms of inflammatory conditions. (Lines 396-397)

We hope the reviewers find our edits have addressed the comments and feedback, and look forward to receiving your responses. Thank you again for consideration of our manuscript for publication in the journal of clinical medicine.

Sincerely,

Hael Abdulrazeq.

Reviewer 2 Report

Comments and Suggestions for Authors

see attached file

Author Response

Dear reviewer and editorial team of JCM,

The authors appreciate the insightful commends and feedback provided by the reviewer, and have made edits to the accompanied manuscript as suggested. The following is a point-by-point response to the comments by the reviewer:

  • As the reviewer notes, the histopathological contents of the filum terminale are not typically factored into surgical selection of patients, rather the clinical picture and the imaging if supportive. The line "must be considered" was revised from the introduction as suggested. (Line 35-37)
  • The word excision has been revised to "transection," which is the more commonly established procedure in the literature. (Line 37)
  • Indeed, not all findings were "pathologies," so we replace that word with just "findings" to keep it broad. (Line 107)
  • The correction is noted, given that we do indeed drill some of the lamina above and below off, we corrected the procedure name in the manuscript to combined inter laminar/translaminar approach (Line  85).

We hope that this revision has addressed the figures. The authors appreciate the valuable feedback and comments.

Sincerely,

Hael Abdulrazeq, MD